# TIcagrelor in Rotational Atherectomy to Reduce TROPonin Enhancement: The TIRATROP Study, A Randomized Controlled Trial

**DOI:** 10.3390/jcm12041445

**Published:** 2023-02-11

**Authors:** Thibault Lhermusier, Pascal Motreff, Vincent Bataille, Guillaume Cayla, Bruno Farah, Jerome Roncalli, Meyer Elbaz, Nicolas Boudou, Fransisco Campello-Parada, Frederic Bouisset, Geraud Souteyrand, Emilie Berard, Vanina Bongard, Didier Carrie

**Affiliations:** 1Hôpital Rangueil, CHU de Toulouse, Université Toulouse 3, 31400 Toulouse, France; 2Département de Cardiologie, Hôpital Gabriel Montpied, CHU Clermont-Ferrand, Université Clermont-Auvergne, 63000 Clermont-Ferrand, France; 3ADIMEP (Association Pour la DIffusion de la MEdecine de Prévention), UMR 1027 INSERM, Université Toulouse 3, 31400 Toulouse, France; 4CHU de Nîmes, 30029 Nîmes, France; 5Clinique Pasteur, 31076 Toulouse, France; 6Département de Cardiologie, Hôpital Gabriel Montpied, CHU Clermont-Ferrand, Université Clermont-Auvergne, CNRS SIGMA UCA UMR 6602, 63000 Clermont-Ferrand, France; 7Département d’Epidémiologie, Economie de la Santé et Santé Publique, Service d’Epidémiologie Centre Hospitalier Universitaire (CHU) de Toulouse, UMR 1027 INSERM, Université Toulouse 3, 31000 Toulouse, France

**Keywords:** ticagrelor, coronary artery disease, rotational atherectomy, clopidogrel, percutaneous coronary intervention

## Abstract

Background: Because rotational atherectomy (RA) is associated with arterial trauma and platelet activation, patients treated with RA may benefit from more potent antiplatelet drugs. The aim of this trial was to assess the superiority of ticagrelor over clopidogrel in reducing post procedure troponin release. Methods: TIRATROP (TIcagrelor in Rotational Atherectomy to reduce TROPonin enhancement) is a multicenter double-blind randomized controlled trial that included 180 patients with severe calcified lesions requiring RA who received either clopidogrel (300 mg loading dose, then 75 mg/d) or ticagrelor (loading dose 180 mg then 90 mg twice daily). Blood samples were collected at the beginning (T0), and 6, 12, 18, 24 and 36 h after the procedure. Primary end point was troponin release within the first 24 h using area under the curve analysis (troponin level as a function of time). Results: The mean age of patients was 76 ± 10 years, 35% had diabetes. RA was used to treat 1, 2 or 3 calcified lesions in 72%, 23% and 5% of patients, respectively. Troponin release within the first 24 h was similar in both the ticagrelor (adjusted mean ±SD of ln AUC 8.85 ± 0.33) and the clopidogrel (8.77 ± 0.34, *p* = 0.60) arms. Independent predictors for troponin enhancement were acute coronary syndrome presentation, renal failure, elevated C-Reactive protein and multiple lesions treated with RA. Conclusion: Troponin release did not differ among treatment arms. Our results suggest that greater platelet inhibition does not affect periprocedural myocardial necrosis in the setting of RA.

## 1. Introduction

Inhibition of P2Y12 receptor combined with aspirin is still the cornerstone therapy for patients undergoing percutaneous coronary intervention (PCI). Three oral P2Y12 inhibitors with substantial pharmacokinetic and pharmacodynamic differences are currently on the market: clopidogrel and prasugrel–second and third generation thienopyridine, and ticagrelor, a non-thienopyridine P2Y12 inhibitor. Ticagrelor is a direct-acting drug that provides faster and greater P2Y12 inhibition than clopidogrel [1]. In contrast to clopidogrel, the effect of ticagrelor on platelet aggregation shows less interindividual variability [2], and its benefits in the long-term management of acute coronary syndrome (ACS) patients was documented in a large pivotal randomized trial [3]. Ticagrelor is currently indicated to reduce the risk of cardiovascular ischemic events in the setting of ACS and is recommended as a first line therapy over clopidogrel in ACS patients [4,5]. Off-label use of ticagrelor is increasing in patients undergoing high risk elective PCI (left main, diabetics, multiple stenting, high risk of stent thrombosis, etc.) but is not supported by scientific evidence [6]. The ALPHEUS (Assessment of Loading With the P2Y12 Inhibitor Ticagrelor or Clopidogrel to Halt Ischemic Events in Patients Undergoing Elective Coronary Stenting) study failed to demonstrate ticagrelor superiority over clopidogrel in reducing periprocedural myocardial necrosis in stable coronary patients undergoing high-risk elective PCI. Data comparing ticagrelor with clopidogrel in stable coronary artery disease (CAD) patients are limited, and this drug has never been specifically evaluated in patients treated with rotational atherectomy (RA) [7,8,9]. RA facilitates PCI for de novo severely calcified or unexpandable lesions, which are increasingly prevalent in the elderly population. RA use remains infrequent, accounting for 3% to 5% in selected high-volume centers, which explains the lack of randomized studies. This historical calcium debulking technique is associated with slow-flow and distal embolization leading to periprocedural creatine-kinase myocardial band (CK-MB) release [10,11,12]. In RA, the interaction between platelets and atheromatous debris is a potential mechanism for cardiac enzyme and troponin release. Indeed, during plaque ablation, the rotating burr might damage the endothelial cell barrier, leading to collagen exposure and platelet recruitment and activation [13,14].

The impact of more potent P2Y12 inhibitors on ischemic complications of RA is unknown. In the TIRATROP study (TIcagrelor in Rotational Atherectomy to reduce TROPonin enhancement; NCT02505399), we assessed whether ticagrelor would be superior to clopidogrel to lower periprocedural troponin release following RA.

## 2. Materials and Methods

### 2.1. Study Design

The TIRATROP study was a multicentric (4 centers across the French metropolitan territory) superiority randomized (1:1) double-blind controlled study that compared ticagrelor and clopidogrel through a parallel group design. Stable CAD patients (stable angina or silent ischemia) or patients presenting with a non ST-elevation ACS with normal troponin level or normal CK-MB level (<3 times the upper limit of the laboratory) at the time of the procedure with at least one de novo highly calcified lesion eligible for RA procedure in a native vessel were screened for inclusion. The main exclusion criteria were ACS with troponin elevation before PCI, patients treated with GpIIbIIIa inhibitors and presence of contraindication to ticagrelor. The complete list of exclusion criteria is presented in online-only Appendix A. All patients gave a written consent for the PCI procedure, and the study protocol was reviewed and approved by the local ethic committee (Comite de Protection des Personnes Sud-Ouest et Outre-Mer).

### 2.2. Antithrombotic Therapy Management and Procedural Technique

All interventions were performed by experienced operators using conventional technique and a standard antithrombotic therapy beside the P2Y12 regimen: aspirin 75 to 160 mg daily started at least the day before the procedure–and continued indefinitely–and parenteral heparin bolus (60–70 UI/kg). Ticagrelor or clopidogrel was administered in a double-blind manner. In the intervention group, ticagrelor was administered orally according to the following scheme: 180 mg on the evening preceding—and at least 6 h before—the index procedure (Day − 1), 90 mg on the morning of the procedure (D Day, prior the index procedure) and 90 mg in the evening (D Day, after the index procedure), and then, 90 mg, twice, on the day following the index procedure (Day + 1). In the control group, clopidogrel was administered orally according to the following scheme: 300 mg on the evening preceding—and at least 6 h before—the index procedure (Day − 1), 75 mg on the morning of the procedure (D Day, prior the index procedure) and 0 mg in the evening (D Day, after the index procedure), and then, 75 mg once, on the day following the index procedure (Day + 1). The maintenance dose of P2Y12 inhibitors was administered until hospital discharge according to randomization. In order to ensure blindness, patients in the intervention group took active pills of ticagrelor and placebo pills of clopidogrel and vice versa for patients in the control group. Differences between active and placebo pills were not distinguishable to the naked eye. Randomization was stratified upon recruitment center and patient’s clinical status (stable or following an ACS). In each stratum, a design based on randomized blocks of 2 to 4 patients was applied according to a 1:1 ratio. After each patient’s inclusion, investigators logged onto a secured website to obtain the number of the corresponding treatment kit following the pre-established randomization list.

The interventional strategy was at the discretion of the physician. RA was performed with a Rotablator coronary system (Rotablator^®^, Boston Scientific Corp., Natick, MA, USA). The smallest burr necessary to create a channel to modify the plaque and to facilitate the delivery of other devices was encouraged. A pressured rotablator-flush solution was infused into the device (1000 UI heparin and 25 mg nitrates in 500 mL saline infusion). Following ablation, balloon predilatation and stent implantation were highly recommended. Baseline clinical, angiographic and procedural characteristics were recorded in a case report form. Discharge treatments were left to the investigator’s choice. A systematic phone call to the patient was scheduled 30 days after the procedure to collect any medical event that may have occurred following hospital discharge. Three study populations were preliminarily defined: (1) the “Intention To Treat” (ITT) population: all patients included in the cohort, with at least 1 blood sample after the procedure, (2) the “Per-Protocole” population: all ITT patients with no major protocol deviation, and (3) the “Safety population”: all patients included who received at least one dose of treatment.

### 2.3. High Sensitivity Troponin

All patients had preprocedural high sensitivity troponin T (HS-TnT) measurement. After the procedure, 5 sets of HS-TnT were measured (H6, H12, H18, H24 and H36). HS-cTnT was measured with the Roche hs-cTnT assay (Roche Diagnostics) [15]. After centrifugation, serum was frozen at −80 °C in each participating center until measurements in the core laboratory (Institut Federatif de Biologie, CHU de Toulouse) [16]. The 99th percentile among healthy subjects is 14 ng/L, with a 10% analytical variation at 13 ng/L [15].

### 2.4. End Points

The primary end point was HS-TnT release within 24 h following the index procedure assessed as an area under the curve (AUC) and corresponding to troponin level (ng/mL) as a function of time (hours). Measurements were computed using the trapezoidal rule with the “pkexamine” Stata’s command, dedicated to pharmacokinetic measures computations. An example of HS-TnT Ln(AUC) calculation is provided in online-only Appendix A. At the margin, some troponin measurements were missing. In these cases, data imputation was used according to the following rules: imputation by the mean of the preceding and the following value (for example, a missing H06 value was imputed by (H00 + H12)/2), last observation carried forward (LOCF) if the last troponin measurements (H24 and H36) were missing (for example, the H18 observed value was carried to H24 for AUC24h computation if both H24 and H36 values were missing). AUC24h was transformed using the natural algorithm, Ln(AUC24h), to reach a Gaussian distribution.

Secondary end points were procedural and in hospital complications in the Safety population. Angiographic success was defined as <20% obstruction after the procedure with Thrombolysis in Myocardial infarction (TIMI) grade III at the end of the procedure. Coronary dissections were defined using the National, Heart, Lung, and Blood Institute (NHLBI) criteria [17]. Coronary perforations were defined by extravasation or free spilling in the pericardium. Stent thrombosis occurrence was also collected. All patients were followed-up for bleeding, vascular and ischemic complications during hospitalization. Bleeding complications were categorized according to BARC classification [18]. Periprocedural myocardial infarction was defined as new pathologic Q waves in ≥2 contiguous leads. MACCE at day 30 from discharge (30-day MACCE) were collected. MACCE were defined as death, myocardial infarction, target vessel revascularization or stroke/transient ischemic attack and were reported in the Safety population.

### 2.5. Statistical Analysis

The number of patients needed in the study (90 patients in each arm) was calculated on the basis of preliminary results from a non-randomized pilot study conducted on 16 patients treated with clopidogrel (n = 9) or ticagrelor (n = 7) who underwent RA in the University Hospital of Toulouse. Data collection, management and analysis were performed by a dedicated coordinating center (Clinical Research Methods Unit of the Toulouse University Hospital–Unite de Soutien Méthodologique a la Recherche Clinique). Statistical analyses were conducted in a blinded fashion.

The nature of all reported clinical events was adjudicated by independent physicians who were not involved in the study. Categorical data are presented as numbers and percentages and continuous data as means and standard deviations (or medians and interquartile ranges when skewed). Student’s *t*-test or non-parametric Mann–Whitney tests were used to compare continuous variables between the two groups, and the chi-square test or Fischer’s exact test were used to compare categorical variables. Ln(AUC24h) were compared using analysis of variance adjusted for stratification factors (recruitment center and clinical status). Factors associated with Ln(AUC24h) were studied in the ITT population. For univariate analyses, relationships between Ln(AUC24h) and the subject’s characteristics were studied using Student’s t-tests or ANOVAs for categorical variables and using Spearman correlation coefficients for continuous variables. Multivariable analyses were performed using stepwise backward multiple linear regression. Underlying assumptions of linear regression (homoskedasticity, normality of residues) were checked. Statistical analyses were performed using Stata Statistical Software, release 10 (StataCorp, College Station, TX, USA). Statistical significance was assumed at *p* value < 0.05.

## 3. Results

From November 2015 to April 2018, a total of 180 patients was randomized in the study. Six patients did not have any blood sample after the index procedure, and one patient under tutorship was mistakenly included. Finally, 84 patients in the clopidogrel group and 89 patients in the ticagrelor group were considered in the ITT analysis (online-only Appendix A). Baseline clinical characteristics were not significantly different between both groups (Table 1). The overall population represented a high-risk population: mean age was 76 ± 10 years, 76.3 % of the patients were men, 35.3% had diabetes and 12.7% were active smokers. Mean left ventricular ejection fraction was 51.5 ± 10 %, and nearly 15 % of the population had severely reduced ejection fraction (≤35%). The initial clinical presentation was stable CAD in 87.3% (n = 151) of the cases and non ST-elevation myocardial infarction with negative biomarker in 12.7% (n = 22).

There were no differences in angiographic and procedural characteristics (Table 2). Three-vessel CAD (or equivalent) was observed in 44% and left main stenosis (lesion > 50%) in 23.7% of the patients.

For the index procedure, the trans-radial approach was widely adopted and concerned 84% of patients. The vast majority of the population had only one calcified lesion involved in the RA procedure. Small burrs (diameter 1.5 mm or less) were mainly selected in 83.5% of cases. The burr-to-artery ratio was 0.46 ± 0.06 mm. A left main stenosis was involved in the RA procedure in 18.1% of patients. Median duration of RA was 36 (24–60) seconds. Stents were implanted in almost all procedures (stent implantation was delayed for two patients, and one patient did not have any stent due to an occlusive dissection without reperfusion). Procedural success was achieved in 97.6% and in 97.8% of patients in the ticagrelor and the clopidogrel group, respectively.

### 3.1. Periprocedural Troponin Release

In the overall population, the incidence of Type 4 MI and myocardial injury according guidelines on the fourth definition of myocardial infarction were, respectively, 72.3% and 99.4%.

As shown in Table 3, troponin release within the first 24 h was comparable in both arms (adjusted mean ± SD of ln AUC 8.77 ± 0.34 in the clopidogrel arm vs. 8.85 ± 0.33 in the ticagrelor arm, *p* = 0.60). No differences were observed concerning troponin release between ticagrelor and clopidogrel groups according to initial diagnosis at admission (Figure 1). Similar results were found in our per protocol analysis (data not shown). Univariate analysis of factors associated with troponin elevation are presented in online-only Appendix A. As shown in Table 4, using multiple linear regression analysis, ACS presentation, renal failure, elevated C-Reactive protein and the presence of multiple lesions treated with RA during the index procedure were independently associated with HS-TnT release.

### 3.2. Periprocedural and In-Hospital Complications

In-hospital cardiovascular events were analyzed in the Safety population (online-only Appendix A). There were two deaths (one in each arm). One patient died from a cardiogenic shock during the procedure, and the other experienced multiple complications (acute heart failure, renal failure, bleeding) leading to death. Only one patient experienced a myocardial infarction in the clopidogrel arm. One major BARC-defined bleed was observed in each group. There were no stent thrombosis events in any of the treatment groups. Few events were observed after hospital discharge without any significant differences between both arms (data not shown).

## 4. Discussion

TIRATROP is the first trial comparing ticagrelor with clopidogrel in patients with calcified lesions treated with RA. We failed to demonstrate the superiority of ticagrelor over clopidogrel to limit the extent of myocardial injury during RA procedures. Indeed, our results suggest that ticagrelor has no superior efficacy in the prevention of ischemic complications of RA, such as no reflow/slow flow. Few data are available about the effect of ticagrelor on post PCI troponin rise in stable CAD patients, and results were controversial until the ALPHEUS trial [7,8,9]. This trial did not show any difference between ticagrelor and clopidogrel in elective PCI on ischemic end points in a high-risk population for ischemic events despite relevant differences in pharmacodynamic effects among both arms. Beside calcified type C lesions, multiple high-risk features were considered as inclusion criteria in ALPHEUS. Considering the presence of calcified lesions as an inclusion criterion, our trial was conducted in a more homogenous population. Our cohort is also associated with a very high-risk for ischemic complications. A greater incidence of transient myocardial ischemia, slow flow/no flow and non-Q-wave myocardial infarction have been reported during RA procedures [19]. Despite this higher ischemic risk, myocardial necrosis was not reduced with ticagrelor’s use in the present study. Beside platelet aggregation, we hypothesize that microembolization of atheromatous debris and thermal injury may also contribute to increase the risk of peri-procedural myocardial injury in RA procedures [20].

GpIIb/IIIa inhibitor use has also been investigated in RA procedures [21,22,23]. According to these studies, GpIIbIIIa blockade is associated with a reduction in post-PCI myocardial necrosis, suggesting that platelet aggregation plays a significant role in ischemic complications occurring after RA. Over the past 20 years, recommendations for atherectomy have evolved towards less aggressive RA procedures in order to reduce ischemic complications [24,25]. In the present study, RA procedures were performed in accordance with modern standards (short procedural time and short ablation duration, avoidance of high burr speed or high burr/artery ratio). A better understanding of the determinants of RA ischemic complications and prevention may lessen the need for more potent anti-thrombotic drugs. Orbital atherectomy and intravascular lithotripsy are also techniques for calcified lesion preparation to facilitate stent implantation. Intravascular lithotripsy is a promising device that could be safer than atherectomy by decreasing atheromatous embolization risk.

High sensitivity cardiac troponin measurement is currently the gold standard to detect myocardial cell damage in the bloodstream following PCI, and procedural myocardial injury is frequently observed after PCI even in stable CAD patients [26,27]. Prognostic significance of troponin release after PCI is still under debate, but meta-analysis and recent data showed a relationship between long-term mortality and post-PCI troponin [28,29]. The predictive value of troponin increases with troponin enhancement, especially if it represents a new increase instead of a natural rise [30]. In the present study, troponin rise was particularly high, reaching the biological definition of Type 4a myocardial infarction in most of these complex cases (elevation of cardiac troponin of over 5 times the 99th percentile upper reference limit for patients with normal baseline values). Troponin release is strictly related to the procedure, as baseline troponin assessment was performed prior to PCI. Consistently, patients with multiple RA-treated lesions had higher troponin release. ACS presentation is also logically found as a predictor for post-PCI troponin elevation. Vessel thrombus burden is probably higher in such settings. Patients with chronic kidney disease and inflammatory reaction have a greater prevalence of elevated cardiac troponin. Reduced clearance and myocardial damage have been suggested as a possible explanation.

The large size of the sheath usually used for the ablative device and the use of a transfemoral approach combined with a longer procedural time have been historically associated with a high risk of bleeding. Although this study had inadequate statistical power to detect clinical differences between ticagrelor and clopidogrel, few in-hospital bleeding events were reported, and no differences between both treatment regimens were observed. Potential explanations could be the large use of radial access with reasonable sheath size and the absence of treatment with GpIIbIIIa inhibitors, as planned in the study protocol.

## 5. Conclusions

In conclusion, this multicenter, randomized trial shows that ticagrelor is not superior to clopidogrel to attenuate troponin release after RA procedures. Moreover, our results suggest that myocardial injury induced by RA is not influenced by the degree of P2Y12 inhibition.

## 6. Study Limitations

The limited size of the cohort may be the cause of the failure to identify clinical differences between clopidogrel and ticagrelor. Larger clinical studies would be of interest, but trials are difficult to implement in such niche procedures. In the present study design, a 300 mg clopidogel loading dose instead of 600 mg should have favored effects of ticagrelor compared with clopidogrel. Despite this dosage, no differences were observed between the clopidogrel and the ticagrelor group in terms of troponin release. Treatment duration according to randomization was limited to hospital stay in the present study. A short treatment period contributes to attenuate clinical discrepancies between arms. One of the main advantages of ticagrelor over clopidogrel is its faster onset of action. As the study includes patients with elective procedures and loading was performed on the evening before the procedure as per study protocol, no conclusions can be drawn on patients needing ad hoc atherectomy. Some protocol deviations were observed, mostly due to the absence of complete adherence to the allocated treatment regimen. However, since the per protocol analysis achieved in 160 patients showed comparable results, we do not believe protocol infringements would have affected final results.

## Figures and Tables

**Figure 1 jcm-12-01445-f001:**
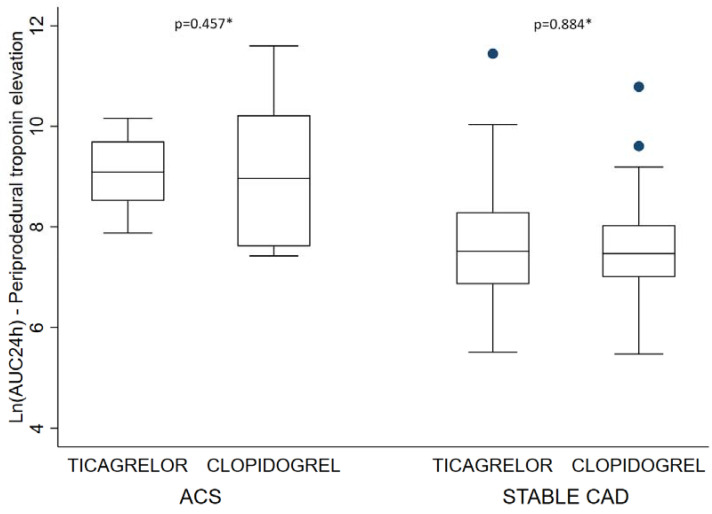
Periprocedural troponin elevation—Ln(AUC24h): * Student’s *t* test; ACS: acute coronary syndrome; AUC: area under the curve; CAD: coronary artery disease.

**Table 1 jcm-12-01445-t001:** Baseline characteristics.

	Clopidogrel (n = 84)		Ticagrelor (n = 89)		*p*-Value
	n	%	n	%	
Diagnosis at admission					0.885
Stable CAD	73	86.9	78	87.6	
ACS	11	13.1	11	12.4	
Male gender	61/84	72.6	71/89	79.8	0.269
Age (years)	76.7	±10.3 *	74.8	±9.9	0.213
Smoking					0.465
No	47	56.0	41	46.6	
Former smoker	27	32.1	35	39.8	
Current smoker	10	11.9	12	13.6	
Treated arterial hypertension	64/84	76.2	61/89	68.5	0.261
Treated dyslipidaemia	45/84	53.6	44/88	50.0	0.639
Type 2 Diabetes mellitus					0.546
No	56	66.7	56	62.9	
Yes (regimen)	4	4.8	8	9.0	
Yes (treated)	24	28.6	25	28.1	
Body mass index (kg/m^2^)	26.2	± 4.2	26.7	±4.4	0.414
Cardiac frequency (bpm)	72.4	±12.2	73.4	±16.6	0.649
GFR < 60 mL/min/1.73 m^2^ †	32/84	38.1	29/88	33.0	0.481
C reactive protein (mg/L)	2.9	(1.3–6.6) ‡	3.0	(1.2–8.0)	0.985
Systolic blood pressure (mmHg)	139	±22	140	±19	0.682
Diastolic blood pressure (mmHg)	73	±15	75	± 12	0.376
Left ventricular ejection fraction (%)	50.2	±12.5	52.6	±11.6	0.265

* Mean ± standard deviation; † computed using MDRD formula; ‡ median (interquartile range); ACS: acute coronary syndrome; CAD: coronary artery disease; GFR: glomerular filtration rate.

**Table 2 jcm-12-01445-t002:** Baseline angiographic characteristics and procedural characteristics.

	Clopidogrel (n = 84)		Ticagrelor (n = 89)		*p*-Value
	n	%	n	%	
Procedural characteristics					
Left main stenosis					0.108
None	56	67.5	53	59.6	
<50%	6	7.2	16	18.0	
>50%	21	25.3	20	22.5	
LAD stenosis					0.986
None	7	8.3	7	7.9	
<50%	7	8.3	7	7.9	
>50%	70	83.3	75	84.3	
CX stenosis					0.925
None	27	32.1	30	33.7	
<50%	11	13.1	10	11.2	
>50%	46	54.8	49	55.1	
RCA stenosis					0.224
None	20	23.8	13	14.6	
<50%	5	6.0	9	10.1	
>50%	59	70.2	67	75.3	
Number of lesions treated with RA					0.497
1	64	76.2	59	67.8	
2	17	20.2	23	26.4	
3	3	3.6	5	5.7	
RA procedure for left main	16/84	19.0	15/87	17.2	0.759
RA procedure for LAD	40/84	47.6	48/87	55.2	0.323
RA procedure for CX	11/84	13.1	14/87	16.1	0.579
RA procedure for RCA	30/84	35.7	32/87	36.8	0.885
Radial vascular access	58/71	81.7	66/76	86.8	0.390
Sheath caliber					0.053
6 Fr	62	87.3	73	96.1	
7 Fr	9	12.7	3	4.0	
Predilatation	81/84	96.4	77/85	90.6	0.124
Stent implantation in RA-treated lesion (s)	84/84	100.0	86/87	98.9	1
PCI for another lesion (same procedure)	30/84	35.7	21/87	24.1	
Number of burrs used					0.659
1	60	72.3	59	67.8	
2	19	22.9	21	24.1	
3 or 4	4	4.8	7	8.0	
Burr diameter *					0.927
1.25	10	11.8	10	11.5	
1.5	57	67.1	60	69.0	
>1.5	18	21.2	17	19.5	
Reference Vessel Diameter (mm) †	3.35	±0.52 ‡	3.34	±0.50	0.908
Total number of burr runs	3.6	±2.0	4.0	±2.3	0.259
Minimal speed used (rev/min)	169,398	±14,140	168,494	±14,158	0.678
Maximal speed used (rev/min)	178,602	±11,210	178,529	±14,268	0.970
Burr runs total length (seconds)	34	(24–56) §	39	(25–62)	0.316
Procedural success	82/84	97.6	87/89	97.8	0.953

* The largest if several burrs used; † the smallest if several lesions treated with RA; ‡ mean ± standard deviation; § median (interquartile range); ACS: Acute coronary syndrome; Cx: left circumflex artery; GFR: glomerular filtration rate; HDL: high density lipoprotein; LAD: left anterior descending artery; LDL: low density lipoprotein; LVEF: left ventricular ejection fraction; PCI: percutaneous coronary intervention; RA: rotational atherectomy; RCA: right coronary artery.

**Table 3 jcm-12-01445-t003:** Periprocedural troponin elevation.

	Clopidogrel (n = 84)		Ticagrelor (n = 89)		*p*
	Mean (SD)	Median (Q1–Q3)	Mean (SD)	Median (Q1–Q3)	
Troponin (ng/L) H00	154 (588)	24 (12–44)	113 (345)	21 (13–42)	0.694
Troponin (ng/L) H06	200 (633)	48 (28–88)	162 (350)	52 (30–130)	0.446
Troponin (ng/L) H12	249 (616)	81 (48–178)	226 (395)	95 (48–225)	0.347
Troponin (ng/L) H18	277 (593)	106 (59–205)	290 (651)	130 (57–286)	0.479
Troponin (ng/L) H24	297 (559)	107 (62–286)	283 (584)	141 (57–298)	0.805
AUC Troponin H00–H24	5746 (14,412)	1847 (1155–4094)	5316 (10,900)	2264 (1108–5275)	
AUC Troponin H00–H24 *(ln)*	7.74 (1.13)	7.52 (7.05–8.32)	7.83 (1.13)	7.72 (7.01–8.57)	
	**Adj. Mean ***	**SD**	**Adj. Mean ***	**SD**	
AUC Troponin H00–H24 *(ln)*	8.77	0.34	8.85	0.33	0.606

* Adjusted for recruitment center and clinical status (stable CAD or Post ACS); ACS: acute coronary syndrome; CAD: coronary artery disease; Q1–Q3: values of 1 and 3 quartiles; SD: standard deviation.

**Table 4 jcm-12-01445-t004:** Multivariate analysis of factors associated with troponin elevation (Ln(AUC24h))—multiple linear regression.

	b	*p*
ACS	1.30	<0.001
GFR > 60 mL/min/1.73 m^2^)	0.52	0.001
C-Reactive protein (mg/L) *	0.13	0.016
>1 lesion treated with RA	0.39	0.021
R^2^ = 0.327		
n = 149		

ACS: acute coronary syndrome; GFR: glomerular filtration rate; RA: rotational atherectomy. * after Ln transformation.

## Data Availability

The datasets generated during and/or analyzed during the current study are available from the corresponding author on reasonable request.

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
