# Peer review of "TIcagrelor in Rotational Atherectomy to Reduce TROPonin Enhancement: The TIRATROP Study, A Randomized Controlled Trial"

_jcm, 2023, doi:10.3390/jcm12041445_

Round 1

Reviewer 1 Report

The paper by Lhermusier and coworkers showed that, according to their multicenter, randomized trial, ticagrelor is not superior to clopidogrel to attenuate troponin release after RA procedures. Moreover, authors suggested that myocardial injury induced by RA is not influenced by the degree of P2Y12 inhibition. The paper is well organized and the results add important information to the field. However I have to point out on some editorial errors that have occurred in this work. 

Minor comments:

1. Page 2, line 46: "thienopyridine–and ticagrelor, a non-thienopyridine P2Y12 inhibitor." In my opinion this sentence should be as follows: "thienopyridine, and ticagrelor, a non-thienopyridine P2Y12 inhibitor."

2. Page 2, line 54: "main, diabetics, multiple stenting, high risk of stent thrombosis…) but is not supported". In my opinion this sentence should be as follows: "main, diabetics, multiple stenting, high risk of stent thrombosis etc) but is not supported"

3. Page 2, lines 63-64: "RA use remains infrequent–accounting for 3% to 5% in selected high-volume 63 centers–which explains the lack of randomized studies". In my opinion this sentence should be as follows: "RA use remains infrequent, accounting for 3% to 5% in selected high-volume 63 centers, which explains the lack of randomized studies".

4. Table 2: the name "Procedural characteristics" should be in the first column. 

5. Page 9, line 290: Please unify "99th" or "99th"

6. Why the authors did not consider to use another P2Y12 inhibitor, cangrelor? Is there any limitation to application of it?

7. Has the sheme of drug administration used by authors been used elsewhere? I mean in clinical routine or scientific reports. If yes, it should be mentioned in the paper.

8. Figure 1: authors should better prepare this graphic - please remove light grey horizontal lines on the level of each value on the OY scale as well as light grey vertical line on the right side of the figure 

Author Response

We thank the reviewer for his suggestions and have changed the manuscript according to comments 1,2,3,4 and 5.

  1. Cangrelor is a novel P2Y12 receptor antagonist that blocks adenosine diphosphate-induced platelet activation and aggregation with unique pharmacodynamic and pharmacokinetic properties. This direct-acting intravenous drug is characterized by immediate onset/offset. We agree with reviewer 1 that rapid onset/offset is an interesting property in the setting of high bleeding/ischemic procedures such as rotational atherectomy. Unfortunately, because this drug is not reimbursed in France, we did not have the possibility to include patients treated with cangrelor.

  1. To the best of our knowledge, the scheme of drug administration has not been used elsewhere.

Reviewer 2 Report

Thank you very much for the opportunity to read an interesting and clinically significant publication summarizing the results of an important study, which compared the effects of clopidogrel with ticagrelor in a group of patients qualified for invasive treatment of ischemic heart disease by rotablation. The results are interesting and provide clinically important information - a similar risk of periprocedural myocardial ischemia characterizes both treatment regimens. Thus, we can probably base the choice of a particular regimen on the clinical characteristics of our patient. After reading, I would like to point out a few minor points to the authors and ask the following questions:
1. Could you please add some information regarding the treatment protocol of the patients? If possible, consider, for example, the recommendations/used GP IIb/IIIa inhibitors and the composition of the fluids used to rinse the system during rotablation. In the center where I work, these issues are often discussed.
2. Referring to point 1, I would like to ask you to specify the information regarding the treatment of patients with GP IIb/IIIa inhibitors. How many patients have received this treatment? What were the drugs? Were there any bail-out passes here (secondary endpoint?)? Was previous treatment with these drugs an exclusion criterion? - such information appears  (line 84), but I do not see it in the description of the study on the clinicaltrials.gov website.
3. The content mentions supplementary materials, but after downloading them, I see only one consort scheme - maybe it's my mistake; please verify it.
4. In the clopidogrel group, all patients had a stent implanted at the rotablation site. In contrast, in the ticagrelor group, three patients did not have a stent implanted after rotablation - do the authors know if the lack of stent implantation affected the observed troponin levels?
5. Two small suggestions - when giving the value of 1 and 3 quartiles, I suggest marking (Q1-Q3); IQR is formally the difference between these values. And the second thing - in table 2 of the baseline characteristic, the distribution of patients with LM stenosis in the clopidogrel group - is one of the patients missing? This is for possible verification.

Author Response

We thank the reviewer for these kind words and hope our revised version will be considered by Journal of Clinical Medicine.

  1. The treatment protocol was mentioned in the manuscript (p 3 line 117).
  2. As mentioned in the study design section of the manuscript, patients treated with GpIIbIIIa inhibitors before the procedure were not included in the present study. Bail-out indications were allowed but none of the patients received GpIIbIIIa inhibitors during the procedure.
  3. We confirm we provided supplementary materials in our initial submission. These files have been uploaded again for resubmission.
  4. We confirm that all patients from the clopidogrel group had a stent implanted. Data were checked concerning the 3 patients in the ticagrelor group. After medical records reviewing, we confirm that only one patient did not have a stent after rotational atherectomy. We sincerely apologize for this mistake and have corrected the Table 2. We do not believe that absence of stent implantation in one patient could have affected the result of the present study.
  5. We changed the interquartile range (IQR) for the values of 1 and 3 quartiles in the manuscript. We confirm that the data is missing concerning the presence of LM stenosis in 1 patient. This patient was treated for LAD lesions but the presence or absence of LM stenosis was not captured.